# Analysis of Strength Development and Soil–Water Characteristics of Rice Husk Ash–Lime Stabilized Soft Soil

**DOI:** 10.3390/ma12233873

**Published:** 2019-11-23

**Authors:** Xunli Jiang, Zhiyi Huang, Fuquan Ma, Xue Luo

**Affiliations:** College of Civil Engineering and Architecture, Zhejiang University, 866 Yuhangtang Road, Hangzhou 310058, China; jxunli@zju.edu.cn (X.J.); hzy@zju.edu.cn (Z.H.); mafuquan@zju.edu.cn (F.M.)

**Keywords:** rice husk ash (RHA), unsaturated stabilized soil, soil–water characteristic curve (SWCC), effective shear strength, scanning electron microscope (SEM)

## Abstract

With increased awareness of environmental protection, the output of traditional curing agents such as cement and lime is less and less, so it is urgent to develop new curing agents with high efficiency and environmental benefits. Thus, this study aims at investigating the application of rice husk ash (RHA) from agricultural waste to the soft soil stabilization. A series of tests are conducted to analyze the strength development process and soil–water characteristics of rice husk ash–lime (RHA–lime) stabilized soils. The results of the strength tests showed that by increasing the content of RHA, the unconfined compressive strength (UCS) and splitting strength of stabilized soils increased first and then decreased. The effective shear strength indexes of the three soil types (soft soil, lime-stabilized soil, and RHA–lime soil) are measured and compared. It is found that RHA can effectively improve the shear resistance and water resistance of stabilized soil. The results of methylene blue test demonstrated that RHA can also promote the reduction of the specific surface area and swelling potential energy of lime-stabilized soil. In addition, the influence of RHA on mineral composition and morphology change in stabilized soils is studied at the microscopic level. The X-ray diffraction tests and scanning electron microscope (SEM) tests showed that strength development and change of soil–water properties of RHA–lime stabilized soil are attributed to enhanced cohesion by cementation and pores filling with agglomerated mineral.

## 1. Introduction

At present, the curing agents commonly used worldwide are still traditional inorganic agents such as cement, lime, and fly ash, etc. However, with increased awareness of environmental protection, the production of cement and lime is affected and the project cost has significantly increased. In order to mitigate the problems, many scholars have studied new types of curing material, which mainly include the industrial and agricultural waste resources in the region where the soil curing project has large engineering volume and high environmental protection requirements. For example, Ta’negonbadi et al. [1] studied the curing effect of lignosulfonate on soils. Ling and Robani [2] studied the curing effect of soft soils with cement and timber industrial ash. Hataf et al. [3] investigated the potential of clay soil stabilization using a biocompatible chitosan solution which was synthesized from shrimp shell waste.

In China, there are many industrial and agricultural wastes in urgent need of resource utilization, among which rice husk ash (RHA) has the widest sources. As the largest rice-growing country in the world, it produces approximately 200 million tons of rice annually, accounting for about one third of the world’s total rice production [4,5]. If all of the rice husks are burned, it can produce approximately 8 million tons of RHA per year. The main chemical substance in RHA is silica, which has high pozzolanic activity under certain combustion conditions [6]. Therefore, if the agricultural waste RHA can be applied in the field of soil stabilization, it has practical significance. It not only increases resource utilization and protects the environment, but also helps reduce the usage of cement and lime. More importantly, its promising curing effect could effectively solve the engineering problems caused by soft soils.

There have been many studies on application of RHA in the field of soil stabilization. Rao et al. [7] studied the effects of different percent of RHA (10%, 20%, 30%, and 40%), gypsum (2%, 3%, 4%, 5%, and 6%), and 5% lime on expansive soils, the results showed that the specimen of 20% RHA, 3% gypsum, and 5% lime provided the highest value of unconfined compressive strength (UCS), 810 kPa after 28 curing days. Based on the studies of Muntohar et al. [8], it was found that when silty soil stabilized by lime, RHA, waste plastic fiber, the highest values of UCS, tensile strength, and California bearing ratio (CBR) were at 4, 5, and 3.6 times more than those of the values for untreated soil, respectively. Rahgozar et al. [9] studied the properties of stabilized clayey sand soil from the Sejzi area with RHA and ordinary Portland cement, the results showed that the UCS and CBR of the 28-day cured specimen with 6% RHA and 8% cement increased by up to 25.44 and 18.2 times more than that of the untreated soil. Ashango and Patra [10] researched the UCS, shear strength, and dynamic properties of stabilized clay soil with different percentages of steel slag, RHA, and lime, and provided the best ratio as: 65% clay, 20% steel slag, 5% lime, and 10% RHA. It was found that the highest values of CBR, UCS, and shear modulus of the optimum stabilized mix design were at 1.975, 1.96, and 1.78 times more than those of the values for untreated soil, respectively. Bagheri et al. [11] presented the results of the consolidated undrained triaxial test and UCS test of treated silty sand soil with cement, lime, and RHA (CLR) admixture, and an increase of the effective cohesion $c^{\prime }$ and effective friction $\phi^{\prime }$ was observed with increasing the CLR content, consistently, and RHA was also found to be effective in increasing the shear strength of CLR-soil mixture. Karatai et al. [12] studied the RHA and natural lime as an alternative to cutting and filling in road construction, and it was found that a combination of 20% by weight of RHA and 2% by weight of natural lime improved the CBR of expansive clay soil by 800%, reduced the soil plasticity by approximately 90%, and decreased free swell by approximately 70%. Based on the studies of Liu et al. [13], the results showed that when the blending ratio of RHA/lime was adopted as 4:1 by weight for soil stabilization, with increase in RHA–lime content and curing time, specific surface area of stabilized expansive soil decreased dramatically and medium particle size increased, deformation properties including swelling potential, swelling pressure, compression index, crack quantity, and fineness of expansive soil lowered remarkably; meanwhile, strength properties involving UCS, cohesion, and internal friction angle improved significantly.

RHA that can stabilize soil is attributed to pozzolanic reaction. When the lime (Ca(OH)_2_) or cement is mixed with RHA, in the presence of water, it releases calcium ions (Ca^2+^) and reacts with the silicates and aluminates to form calcium silicate hydrate (CSH). According to the research of James and Rao [14], CaO_1.5–2.0_SiO_2_·(H_2_O)_1.0–2.5_ (CSHI) is the main product of the reaction between Ca(OH)_2_ and RHA, but Cook [15] questioned the research results, and through study found that CaO_0.8–1.5_SiO_2_2(H_2_O) (CSHII) is the main product. With the deepening of the research, most scholars think that the reaction products of RHA with lime or cement mainly include CSHI and CSHII, which are related to reaction conditions [16]. In the stabilized soils, RHA also can react with soil produce minerals other than CSHI and CSHII. These minerals are mainly related to the type of soils. In order to study the microscopic mechanism of RHA stabilized soils, X-ray diffraction and SEM methods are generally used for analysis. For example, Ali et al. [17] studied lime–RHA mixtures and lime–RHA-residual soil mixtures by X-ray diffraction method, respectively. It was found that the reaction product of lime–RHA mixture is crystallized CSHI, the reaction products of lime–RHA-residual soil mixtures for the strength development are crystallized CSHI, tetracalcium aluminate-13-hydrate (C_4_AH_13_), and Stratlings’s compound (C_2_ASH_8_). Muntohar [18] used X-ray diffraction method to study the mineral composition of four kinds of stabilized soils (including 6% lime, 6% lime + 3% RHA, 6% lime + 6% RHA, and 6% lime + 12% RHA). The results showed that CSH gel and calcium aluminate gel (C_3_AS_n_H_n−2_) were formed in the RHA–lime stabilized soils. Basha et al. [19] analyzed the mineral composition and micro morphology change of RHA–cement stabilized residual soil. The results showed that the addition of RHA causes pozzolanic reaction and new mineral structure in stabilized soil. Rahgozar et al. [9] also studied the micro morphology change of RHA–cement stabilized clayey sand soil by SEM, and it was found that the aforementioned stabilized soil was characterized as a well-structured soil matrix with very small pores, which can be attributed to the pozzolanic reactions of the cement and RHA. 

Previous studies have proved that RHA has a good curing effect on soils when combined with cement, lime, etc. However, there are few studies on the curing effect of RHA on soft soil, especially soil–water characteristics after stabilization. In engineering practice, the soft soil after stabilization is mostly in an unsaturated state, such as excavation foundation pits, fill roadbeds, and earth-rock dams. With the change in climatic environment and external conditions, the water content in the soil changes constantly, which leads to a wider range of negative pore water pressure (suction) and makes the engineering performance more complex than for saturated soil. Therefore, in addition to studying the curing effect of RHA on soft soil, it is necessary to gain a better understanding of the soil–water characteristics of RHA–lime stabilized soil.

The objective of this study is to investigate the effects of RHA and lime stabilization on the strength and soil–water characteristics of soft soil. The paper is organized as follows. The next section introduces the materials and test procedures, followed by the analysis of UCS and split strength of RHA–lime stabilized soil under different conditions and curing times to determine the optimum RHA content. In the subsequent section, the soil–water characteristics of RHA–lime stabilized soil are studied, including the soil–water characteristic curve (SWCC), shear strength in the unsaturated state, and expansion characteristics after water intrusion. Next, the reasons for the increase in strength and the change in soil–water characteristics of the RHA–lime stabilized soil are analyzed by conducting microcosmic tests. The final section summarizes the findings of this work.

## 2. Materials and Laboratory Tests 

### 2.1. Materials

The test soils were taken from a construction site in Hangzhou Metro in China. It is silt clay (a typical soft soil) and the depth of the soil is more than 2 meters. Basic physical properties of the soil were determined according to “test methods of soils for highway engineering” (JTG D40-2007, China) [20], the results are listed in Table 1. The maximum dry density and optimum moisture content were obtained using the light compaction test in this specification [20]. Combining XRD with ESEM-EDAX analysis, it can be found that the soil mainly contained clay minerals of illite, chlorite, and kaolinite, and other minerals such as quartz and muscovite. The principal mineral composition of the soil is marked on the X-ray diffraction pattern shown in Figure 1. According to the Unified Soil Classification System (ASTM D2487-2017) [21], the soil is classified as lean clay (CL) type. Two stabilization materials, lime and RHA, were used in this study. The chemical composition is shown in Table 2. 

The RHA is black powder. The particle size of the RHA was analyzed by Mastresizer 2000 laser particle size analyzer (Malvern Instruments Ltd, Malvern, UK). The results show that the average particle size is 14.924 µm and the specific surface area is 0.402 m^2^/g. The particle size distribution is shown in Figure 2. The microstructure of the RHA was observed by SEM and energy spectrum analysis, as shown in Figure 3. The energy spectrum analysis indicates that the RHA is composed mainly of silicon and oxygen. 

### 2.2. Test Design

Three different mixtures of the soil with lime (3%, 5%, and 7% by weight) were prepared, and different percentages of the RHA (0%, 2%, 4%, 6%, 8%, and 10%) were added. The percentage of lime and RHA content is the weight of lime and RHA relative to the dry soil weight. The tests included the UCS test, indirect tensile test (splitting strength test), SWCC test, shear strength test, methylene blue test, X-ray diffraction test, and SEM. 

### 2.3. Test Methods

#### 2.3.1. Preparation of Specimens

The specimen fabrication was conducted in accordance with the “test methods of materials stabilized with inorganic binders for highway engineering” (JTG E51 2009, China) [22]. First, the soil was air-dried, crushed into powder, and passed through a 4.75 mm sieve. Then, the specimens were compacted under the optimum moisture content, and the degree of compaction was controlled at 95%. Finally, the specimens were demolded and transferred to the curing room with a temperature of 20 ± 1 °C and humidity of 95% ± 5%. The dimensions of the specimen for different tests were: 50 mm in diameter and 50 mm high for the UCS and splitting strength tests, 70 mm in diameter and 40 mm high for the SWCC test, and 61.8 mm in diameter and 20 mm high for the shear strength test.

#### 2.3.2. Strength Test

The UCS test and splitting strength test were conducted in accordance with the JTG E51 2009 China [22]. The tests were conducted using a universal testing machine with an axial strain rate of 1 mm/min. 

#### 2.3.3. Soil Water Characteristic Curve Test 

The SWCC test was measured by the filter paper test specified in ASTM D5298 [23]. Whatman (Maidstone, UK) No. 42 filter paper with a diameter of 55 mm was used in the test. For each type of soil, ten different levels of moisture content were produced.

#### 2.3.4. Shear Strength Test

The shear strength index was determined by the direct shear test. The test was performed in accordance with the standard specification [20]. Six water contents were selected, and for each water content, four specimens were prepared. The shear strength test was conducted at four different stress levels: 50, 100, 150, and 200 kPa.

#### 2.3.5. Methylene Blue Test

The methylene blue test was conducted based on the French Norme Française NF P 94-068 (AFNOR 1993) standard. Methylene blue solution was made by dissolving methylene blue powder in distilled water. The test procedure is the same as that described in Turkoz and Tosun’s paper [24], and the specific process is depicted in Figure 4.

#### 2.3.6. Microscopic Characteristic Test

The stabilized soil was analyzed by PANalytical X’Pert PRO diffractometer (PANalytical B.V., Almelo, The Netherlands). The test used a continuous scan measurement with a scan range of 5–80° and a scan speed of 4 °/min. The SEM used was FEIQuanta 650 FEG (FEI., Hillsboro, OR, USA) with an acceleration voltage range of 200 V–30 kV and maximum beam current of 200 nA. After the compression test, all tested specimens were submerged into anhydrous ethyl alcohol for 1 day and oven-dried at 70 °C for 24 h to terminate the hydration process. The samples used in the SEM test were carefully removed from unbroken sections of the specimen, and had a volume of approximately 1 cm^3^.

## 3. Analysis of Characteristics of Strength Development

### 3.1. Effect of RHA on Unconfined Compressive Strength

The effect of the RHA on the increase of the strength of the lime soil was studied by conducting the UCS test. From the results in Figure 5, Figure 6 and Figure 7, it is obvious that the addition of the lime and RHA as well as curing time have significant impacts on the UCS of the specimens. Under the same dosage, the strength of RHA–lime soil increases with the increase of the curing time. For example, in the 7% lime group, the strength of the stabilized soil (7% lime + 6% RHA) is 1.17, 1.59, and 2.45 MPa at 7, 14, and 28 days of curing time, respectively. It can also be seen from these figures that, under the same curing age, the UCS increases first and then decreases with the increase of RHA content, that is, there is an optimal RHA content. For example, at the curing time of 28 days, the optimum RHA content is 2%, 4%, and 6%, respectively, when the lime content is 3%, 5%, and 7%. Compared with pure lime-stabilized soils, the compressive strength increases by 33%, 25%, and 28% when the lime content is 3%, 5%, and 7%, respectively. 

Rahgozar et al. [9], and Ashango and Patra [10] reported a similar trend that RHA can increase the UCS of stabilized soil, and there is an optimum RHA content. Ali et al. [17] also found that the optimum amount of RHA varies with the amount of lime; this is consistent with the conclusion of our study result. Analysis of the reasons for these phenomena, mainly because the RHA is needed to react with lime to form CSH gelling. When the RHA is increased continuously, the lime that can react with it becomes insufficient. When more lime is in the soil, the amount of RHA that can be reacted increases, that is, the optimum amount of RHA will increase. In addition, the RHA used in this test is the product of industrial boiler combustion; its ash contains incomplete combustion carbon and other impurities. These substances have a certain adverse impact on soil stabilization strength [25]. Thus, excessive increase of RHA will reduce the strength of stabilized soil. According to the study by Boating and Skeete [16], the reaction equation for RHA and lime is as follows:(1)$$CaO+H_{2}O\to Ca{\left(OH\right)}_{2},$$
(2)$$SiO_{2}+Ca{\left(OH\right)}_{2}\overset{}{\to }CSHΙ+CSH\Pi ,$$
in which:(3)$$CSHΙ=CaO_{1.5-2}SiO_{2}{\left(H_{2}O\right)}_{1.0-2.5},$$
(4)$$CSH\Pi =CaO_{0.8-1.5}SiO_{2}2(H_{2}O).$$

### 3.2. Effect of RHA on Splitting Strength

Figure 8 shows the relationship between the splitting strength of 28 days and the content of the RHA and lime. It can be seen that the addition of the RHA has a significant effect on the splitting strength of the stabilized soil. The relationship between splitting strength and RHA content is similar to the relationship between compressive strength and RHA content. Both showed a trend of increasing first and then decreasing, and the optimum RHA content is also the same, 2%, 4%, and 6% respectively. In the 3%, 5%, and 7% lime content groups, the splitting strength of stabilized soils with the optimum RHA content increased by 73%, 52%, and 46%, respectively, compared with that of pure lime-stabilized soil. Compared with the maximum increase values (33%, 25%, and 28%) in the UCS test, the split strength increases more obviously. 

Similar trends were also observed by Yu and Zhao [26] and Muntohar et al. [8] about the effects of RHA on the UCS and splitting tensile strength. This is mainly due to the reaction of the RHA and lime to produce more CSH gel, which acts as cementation, making the soil particles connect better and enhancing the tensile strength. However, the increase of the UCS requires not only cementitious materials to connect soil particles, but also enough materials to fill pores. Therefore, the influence of the RHA on the split strength is higher than that on UCS.

## 4. Analysis of Soil–Water Characteristics

In order to study the soil–water characteristics of stabilized soils, the RHA–lime stabilized soil (5% lime + 4% RHA), lime-stabilized soil (5% lime), and soft soil under 28 days curing time are selected for a comparative analysis. 

### 4.1. Effect of RHA on SWCC

The matrix suction of soft soil, lime-stabilized soil, and RHA–lime stabilized soil with different water contents is determined by the filter paper method. The SWCC model proposed by Fredlund and Xing (1994) [27] is used to fit the test data. The Frendlund–Xing equation is as follows:(5)$$S=C(h_{m})\times \left[\frac{1}{{\left\{\ln\left[\exp(1)+{\left(\frac{h_{m}}{a_{f}}\right)}^{b_{f}}\right]\right\}}^{c_{f}}}\right],$$
where $C(h_{m})$ is the correction factor, defined as:(6)$$C(h_{m})=\left[1-\frac{\ln(1+\frac{h_{m}}{h_{r}})}{\ln(1+\frac{10^{6}}{h_{r}})}\right],$$
where S is the degree of saturation; $h_{m}$ is the matrix suction; $h_{r}$, $a_{f}$, $b_{f}$, $c_{f}$ are model coefficients. The $a_{f}$ value is related to the air entry value (AEV), the $b_{f}$ and $c_{f}$ values are related to the pore size distribution of the soil and the overall symmetry of the SWCC. The SWCC with a large $b_{f}$ value produces a sharp transition near the AEV [28]. 

The effects of RHA and lime on SWCC of soft soil are shown in Figure 9, which is a diagram of the relationship between the saturation and matrix suction. The points in the figure are the test data points, and the curve is the fitting curve. It should be noted that the best fitting parameters and fitting curves are produced by Origin fitting software. The values of the model fitting parameters of three kinds of soils are given in Table 3. The predicted SWCC curves demonstrated a good relationship with the measured SWCC data for both untreated and treated soils. 

As can be seen from the curve in Figure 9, the RHA and lime had significant influence on the SWCC of soft soils. To be more specific, comparing the slopes of the curves of the three soils between the AEV and the residual water content, it can be observed that the slope of the RHA–lime stabilized soil is the largest, followed by lime-stabilized soil and soft soil. This is consistent with the change of $b_{f}$ value in Table 3. The larger the value of $b_{f}$, the more uniform the pore size distribution of the soil. Therefore, this phenomenon indicates that the incorporation of the RHA makes the pore size distribution of the stabilized soil more uniform.

Table 3 displays the variations of the AEV of the three soils. The higher the AEV, the more difficult it is for air to enter the soil, that is, the smaller the pore size, the higher the compactness, the better the water retention capacity. To better understand the fluctuation of the AEV, it is compared and discussed in detail. For example, the AEV of soft soil is approximately 2000 kPa, lime-stabilized soil is approximately 3059 kPa, and the addition of 4% RHA into lime-stabilized soil resulted in an AEV of 7980 kPa. The increase in the AEV of RHA–lime stabilized soil is approximately 295% and 157% compared to that of soft soil and lime-stabilized soil, respectively. This indicates that the addition of RHA makes the stabilized soil more compact and increases the water retention capacity. 

Through the analysis of the SWCC, it can be concluded that the addition of the RHA will make the pore size distribution of stabilized soil more uniform and compact, and increases the water retention capacity. These results are similar to those of the earlier studies on lime-stabilized soil [29,30,31]. The main reason for this phenomenon may be that the flocculation and pozzolanic reaction between RHA and lime can help change the microstructure and interface properties of soil, thus make the pore size distribution more uniform and enhancing the water retention performance of stabilized soil [29].

### 4.2. The Change of Shear Strength with Water Content

Shear strength is an important performance indicator for soils in engineering applications. For unsaturated soils, the change in the shear strength is quite different from that of saturated soils. Therefore, this study uses the SWCC combined with direct shear tests to explore the effect of water content on the shear strength index of unsaturated RHA–lime stabilized soil [32].

The shear strength indexes of three kinds of soil under different water content conditions are measured by the direct shear test. Figure 10 shows the relationship between shear strength and vertical pressure. And Figure 11 shows the relationship between the shear strength index and water content. It can be seen from the figures that the cohesion c is greatly affected by the water content, and the internal friction angle φ is less affected by the water content. Comparing the three kinds of soil, it can be observed that when the water content is low, c of the three kinds of soil has little difference. However, with the increase of the water content, the c of the three soils varies greatly. c of soft soil decreases with the increase of water content, and finally approaches zero. c of lime-stabilized soil and RHA–lime stabilized soil increases first and then decreases, and remains at a certain value under the condition of high water content, among them, the RHA–lime stabilized soil is larger than the lime-stabilized soil. φ of the three soils show a downward trend, and the values of RHA–lime stabilized soil is largest, followed by lime-stabilized soil and soft soil. 

In order to further analyze the change of effective shear strength indexes with water content, the effective shear strength indexes under different water contents are calculated by Equation (7) and the SWCC results. Equation (7) is as follows:(7)$$\tau =c^{\prime }+(\sigma_{n}-\theta fh_{m})\tan\varphi^{\prime },$$
where τ is the shear stress; $\sigma_{n}$ is the vertical stress; $c^{\prime }$ and $\phi^{\prime }$ are the effective cohesion and effective friction angle, respectively; θ is the volumetric water content; f is the saturation factor; $h_{m}$ is the matric suction.

The relationship between the effective shear strength index and the water content of the three soils is shown in Figure 12. By analyzing the relationship between $c^{\prime }$ and the water content of the three soils, it can be seen that for soft soil and lime-stabilized soil, $c^{\prime }$ decreases with the increase of the water content. For RHA–lime stabilized soil, the $c^{\prime }$ initially is improved by increasing the water from 10% to 12% and then reduced when the water from 12% to 21%. Under the condition of low water content, $c^{\prime }$ of the three soils differs greatly, among which RHA–lime stabilized soil is the largest. Under the condition of high water content, $c^{\prime }$ of lime-stabilized soil and soft soil gradually approach each other and eventually converge. This shows that the water stability of lime-stabilized soil is poor. Thus, the addition of the RHA to lime-stabilized soil can greatly enhance $c^{\prime }$, namely effectively improving the water resistance of the soil. Effective internal friction angle $\phi^{\prime }$ is similar to internal friction angle φ, the values of RHA–lime stabilized soil is largest, followed by lime-stabilized soil and soft soil.

Bagheri et al. [11] and Liu et al. [13] also observed similar trends, that is, RHA can effectively improve the shear strength index of stabilized soil. The reason may be that the active silica in RHA promotes the hydration reaction in lime-stabilized soil, and the resulting cementitious material fills the voids of clay, which leads to the reduction of pore size and a more uniform pore size distribution. This leads to an increase in the matric suction, which in turn increases the effective cohesion. And cementitious material also makes the connection between the soil particles better, thereby increasing the effective internal friction angle. 

### 4.3. Methylene Blue Test Analysis

To analyze the effect of the RHA on the expansion characteristics of the soil when water intrudes, a methylene blue test was conducted. The specific surface area and methylene blue value (MBV) of the soil particles were measured using the methylene blue test. The formula for calculating the specific surface area of soil particles is shown in Equation (8) [33], and the MBV is computed by Equation (9) [34]:(8)$$SSA=\frac{m_{MB}}{319.87}A_{V}A_{MB}\frac{1}{m_{s}},$$
where SSA is the specific surface area of the soil particles; $m_{MB}$ is the mass of the adsorbed methylene blue at the point of complete cation replacement,$A_{V}$ is Avogadro’s number (6.02 × 10^23^/mol), $A_{MB}$ is the area a covered by one methylene blue molecule (typically assumed to be 130 ${\overset{{}^{\circ}}{A}}^{2}$), and $m_{s}$ is the mass of the soil specimen.
(9)$$MBV(g/100g)=\frac{V_{cc}}{m_{s}}*100*c_{cc},$$
where MBV is the methylene blue value, $V_{cc}$ is the volume of the methylene blue solution injected into the soil solution, and $c_{cc}$ is the methylene blue solution concentration.

The methylene blue test results are shown in Table 4. By comparing the three soils, it can be observed that soft soil has the largest surface area, lime-stabilized soil is smaller, and RHA–lime stabilized soil is smallest. This indicates that the specific surface area of the soft soil particles after stabilization is significantly reduced, and the incorporation of the RHA will promote the reduction of the specific surface area of the lime-stabilized soil. 

Liu et al. [13] reported a similar trend, which also found that with increase in RHA–lime content and curing time, specific surface area of stabilized expansive soil decreased dramatically and medium particle size increased. This is mainly because the hydrated calcium silicate gel formed by the reaction of RHA and lime, which enhancing the soil agglomeration effect.

According to the methylene blue test results, the expansion characteristics of the soil can also be analyzed. Yukselen and Kaya [35] proposed the analysis of soil expansion characteristics based on the methylene blue value, and gave a judgment table, as shown in Table 5. According to the judgment table, the degree of expansion of the soil can be determined.

It can be observed from the judgment table that when the MBV is larger, the degree of expansion of the soil is larger, that is, the swelling potential is larger. Comparing the MBV of the three soil samples, the soft soil obtained in this experiment has an MBV of 1.23 (g/100 g), which indicates a low-expansion soil. The MBV of the soil after stabilization is reduced, especially after the incorporation of the RHA. The MBV decreases to a greater extent, which means that the addition of the RHA causes the soil swelling potential to decrease more. This is consistent with the findings of Muntohar [36].

## 5. Microscopic Tests Analysis

### 5.1. XRD Phase Analysis

In order to investigate the effects of the RHA on the formation of chemical products and the change of phase characteristics in lime-stabilized soil, X-ray diffraction analysis was conducted on these two stabilized soils. The samples selected for analysis were lime-stabilized soil (5% lime) and RHA–lime stabilized soil (5% lime + 4% RHA) under 28 days curing time. The diffraction pattern is shown in Figure 13, where 1 indicates lime-stabilized soil and 2 indicates RHA–lime stabilized soil.

It can be seen from Figure 13 that the diffraction patterns of the lime-stabilized soil and the RHA–lime stabilized soil are basically the same. The mineral composition did not changed. There are quartz, muscovite, calcite, CSH gel, and other substances, which indicates that the RHA has little effect on the crystallized primary mineral composition of the soil-stabilizer mixture. The intensity of the diffraction peaks of some of the corresponding substances are slightly different, for example, the peak of CSH gel of RHA–lime stabilized soil is higher than lime-stabilized soil. This is in line with Muntohar’s study [18]. The difference is that calcium aluminate gel (C_3_AS_n_H_n-2_) is also present in the analysis of Muntohar. The reason may be due to the difference in the chemical composition of the soils and the activity of the stabilized materials. 

### 5.2. SEM Test Analysis

The microstructure is the basis of physical and mechanical properties of the soil, which largely determines the macroscopic strength of the soil. In order to better analyze the reasons for the increase in strength and the change in soil-water characteristics of the RHA–lime stabilized soil, several representative samples were selected for comparison, and the magnification was observed to be 2000 times and 5000 times. First, the microstructural changes of soils with different mix ratios under the same curing age were analyzed. Here, the soft soil, the lime-stabilized soil (5% lime), and the RHA–lime stabilized soil (5% lime + 4% RHA) specimens under 28 d curing time were selected for comparative analysis. Then, the microstructural changes of stabilized soils with the same mix ratio at different curing time were analyzed. Here, the RHA–lime stabilized soil (5% lime + 4% RHA) under 7, 14, and 28 days curing time were selected.

Figure 14a1,a2 exhibits the SEM image of soft soil. As can be seen from these images, the microstructure of the soft soil is composed of a large volume of agglomerated structure, flaky structure, and block structure, and the overlap forms are mostly surface to surface, surface to edge, and edge and edge. Moreover, because of the lack of the cementitious products, more voids are visible. Figure 14b1,b2,c1,c2 shows the SEM images of lime-stabilized soil and RHA–lime stabilized soil after curing of 28 d. By observing at 5000 times magnification, the connection mode of the stabilized soil structure and the crystal growth between the pores can be clearly observed. In lime-stabilized soil microstructure, there is a certain amount of acicular hydrated calcium silicate gel formed and its shape is mostly short and thin, partially overlapped. This result is consistent with the study of Kassim and Chern [37], which observed the presence of needle-like crystals in lime soil. In the RHA–lime stabilized soil microstructure, the hydrated calcium silicate gel is produced in a large amount, mostly acicular and columnar crystals, and its shape is thicker and longer. The crystals are aggregated to form a bundle, with pores filled and overlapped more with other crystals. Comparison of Figure 14b1,b2,c1,c2 illustrates great change in the lime soil structure due to the addition of RHA. It is obvious that the mineral structure of the lime stabilized soil has changed from a thin acicular form to a more agglomerated columnar mixture since the RHA promotes pozzolanic reaction. 

Compared with the study of Yu et al. [38], there is a certain difference in the morphology of the CSH gel. They observed that the CSH gel obtained by the reaction of Ca(OH)_2_ with RHA is appears to be flocs in morphology, with a porous structure and large N_2_-specific surface. This is different from what this paper observed. Analysis of the reasons are mainly due to the difference in reaction conditions and raw materials. Yu et al. only analyzed the reaction between RHA and Ca(OH)_2_, and the reaction conditions were carried out in a 40 °C saturated Ca(OH)_2_ solution to finally obtain CSH gel. According to the study by Taylor [39], CSH gel exhibits a different structure under different formation conditions. Under normal temperature conditions, it mainly appears as four basic forms of network-like particles, equal-grained particles, fibrous particles, and internal products. Therefore, the fibrous and columnar minerals in the RHA–lime stabilized soils observed in this paper are also reasonable. This further illustrates that the reaction products of RHA and lime have a great relationship with the reaction conditions.

The SEM images of the RHA–lime stabilized soil samples at 7, 14, and 28 days of curing time are shown in Figure 15. Under the condition of 2000 times, it can be found that the pore content in the microstructure of RHA–lime soil decreased with the increase of curing age. Under the condition of 5000 times, it can be observed that with the increase of curing time, the number of acicular minerals in the stabilized soil is increasing, characterized by lengthening, thickening, and agglomeration. Under the condition of 7 d, some acicular minerals appear between the pores in the RHA–lime stabilized soil, but they are mostly scattered and slender. Under the condition of 14 days, more acicular and columnar minerals appear in the RHA–lime stabilized soil, with localized mineral agglomeration and filling of pores. Under the condition of 28 days, the minerals in the RHA–lime stabilized soil are mainly columnar minerals. The amount of the minerals is large, the shape is long and thick, the bundles are agglomerated to fill the pores, and overlapped well with others. The analysis illustrates that the minerals in the RHA–lime stabilized soil will continue to develop, it gradually becomes thick from the slender shape, and finally aggregates to fill the pores when the curing time increases.

The SEM results agree with the stabilization mechanism [14,15,16] for the RHA–lime stabilized soil. Based on their micro-chemical analysis, when mixture of water, lime, and RHA is added to the soil, water decomposes into hydroxyl ions (OH^−^) and hydrogen (H^+^), and then reacts with calcium oxide to form Ca(OH)_2_. In the presence of a large amount of hydroxide, the active silica in the RHA will continue to react with Ca(OH)_2_ to form hydrated calcium silicate gelling substances CSHI and CSHII. These products fill the pores of the soil, making the pores smaller and denser in the structure. Slender acicular and agglomerated columnar minerals make the connection between the particles and the particles more compact. Therefore, based on the SEM results, the strength increased and soil–water characteristics change with the addition of the RHA is attributed to enhanced cohesion by cementation and pore fillings by agglomerated minerals. This further confirms the correctness of the discussion of the previous test results.

## 6. Conclusion

In this research, the effects of the rice husk ash (RHA) on the stabilization of soft soil are investigated in terms of strength development and soil–water characteristics. The findings of the laboratory investigation can be summarized as following cases:The RHA has a significant effect on the unconfined compressive strength and splitting strength of the lime-stabilized soil, of which the impact on the latter is greater. The optimum dosage of the RHA is 2%, 4%, and 6% when the lime content is 3%, 5%, and 7%, respectively.The RHA has a great influence on the soil–water characteristic curve of lime-stabilized soil. After adding RHA, the air entry value of the soil–water characteristic curve increases obviously, and the curve between the air entry value and residual water content is steeper. It demonstrates that the RHA makes the pore size distribution more uniform and the soil structure more compact.With the increase of the water content, the effective cohesion of the RHA–lime stabilized soil increases first and then decreases, which is always greater than that of lime-stabilized soil and soft soil. Especially in the state of high water content, the RHA can enhance the water stability of lime-stabilized soil.The results of the methylene blue test show that the incorporation of the RHA promotes the reduction of the specific surface area and swelling potential of lime-stabilized soil.Through the scanning electron microscope test, the mineral structure of the lime-stabilized soil changes from a thin acicular to form a more agglomerated columnar mixture when the RHA is added. As the curing time increases, the cementitious minerals in the RHA–lime stabilized soil increase continuously, and the overlap between them is more obvious.The microscopic test analyses reveal that the main reasons for strength increase and soil–water characteristics change with the addition of RHA are the enhanced cohesion by cementation and pore fillings with agglomerated minerals.

## Figures and Tables

**Figure 1 materials-12-03873-f001:**
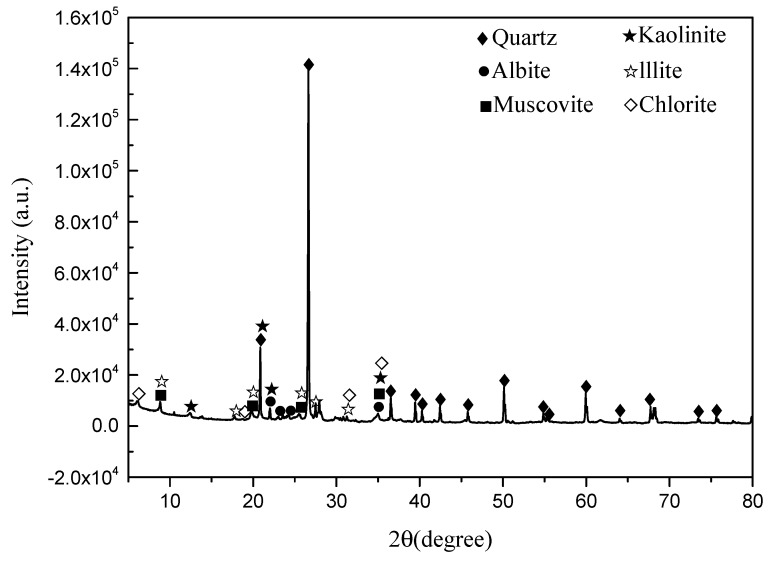
X-ray diffraction pattern of soil.

**Figure 2 materials-12-03873-f002:**
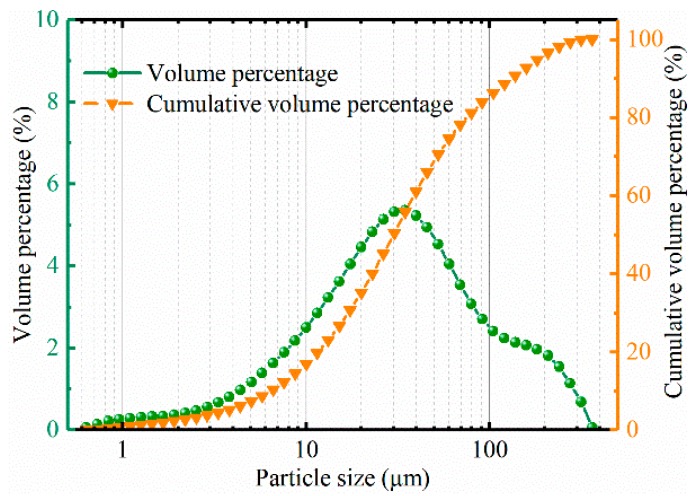
Particle size distribution of rice husk ash.

**Figure 3 materials-12-03873-f003:**
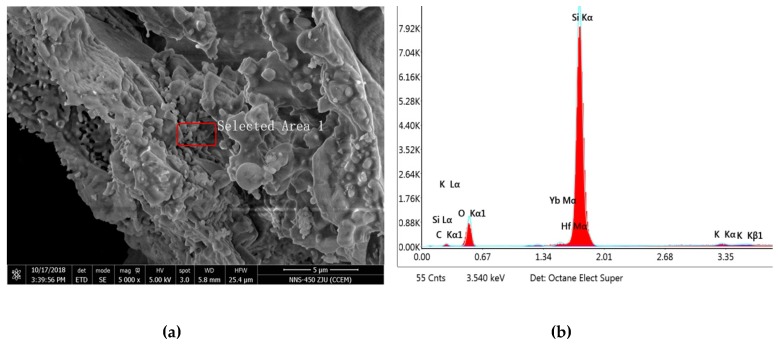
SEM diagram (**a**) and energy spectrum analysis (**b**) of the RHA (rice husk ash).

**Figure 4 materials-12-03873-f004:**
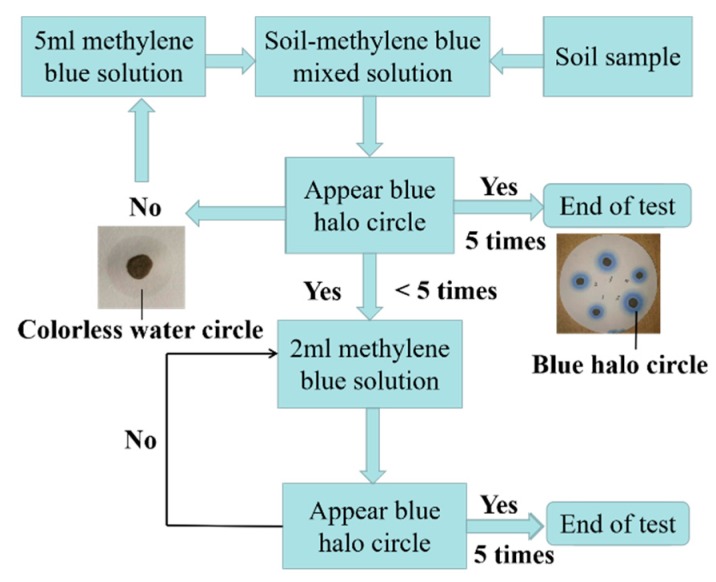
Methylene blue test flow chart.

**Figure 5 materials-12-03873-f005:**
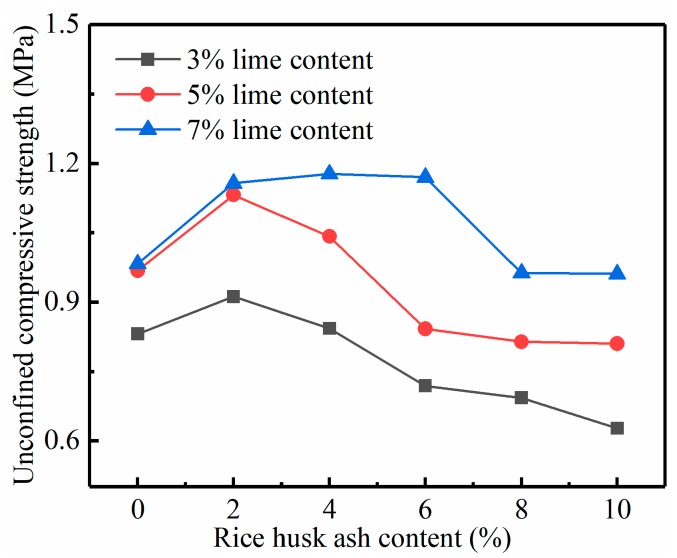
Relationship between the UCS (unconfined compressive strength) and RHA content in 7 days.

**Figure 6 materials-12-03873-f006:**
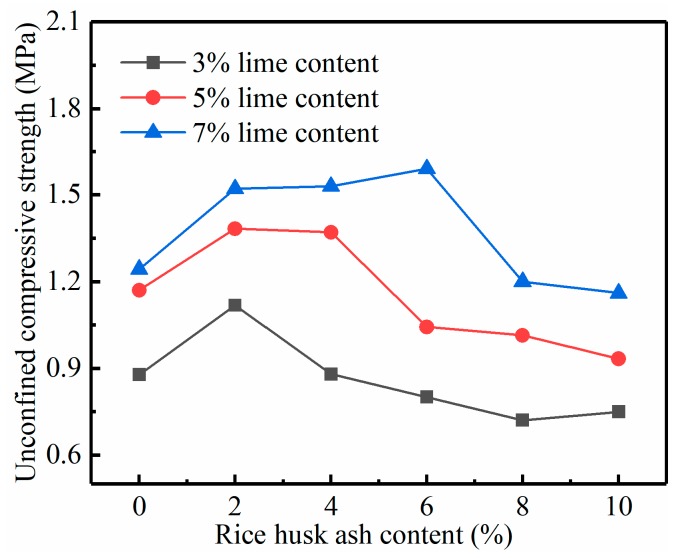
Relationship between the UCS and RHA content in 14 days.

**Figure 7 materials-12-03873-f007:**
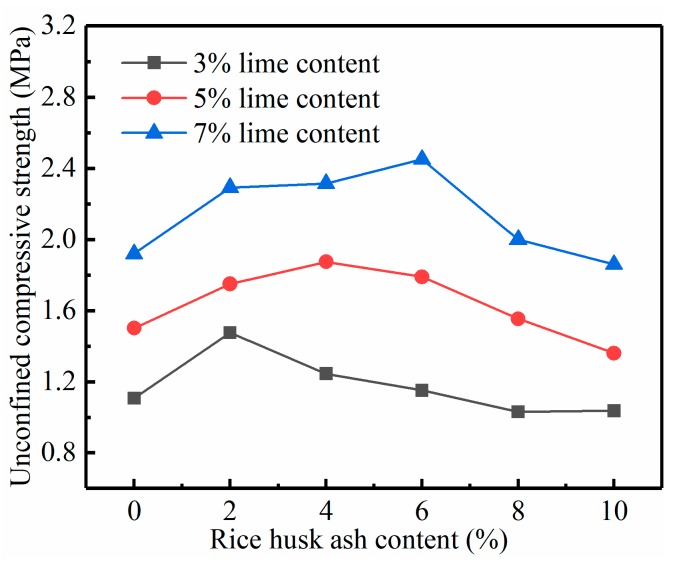
Relationship between the UCS and RHA content in 28 days.

**Figure 8 materials-12-03873-f008:**
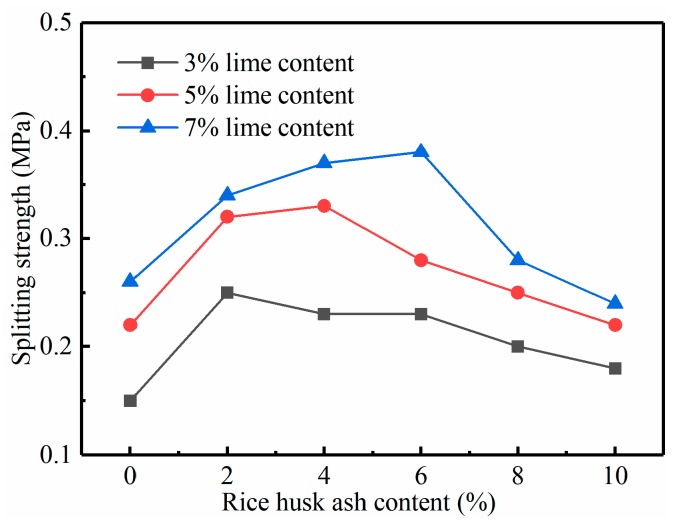
Relationship between splitting strength and content of the RHA in 28 day.

**Figure 9 materials-12-03873-f009:**
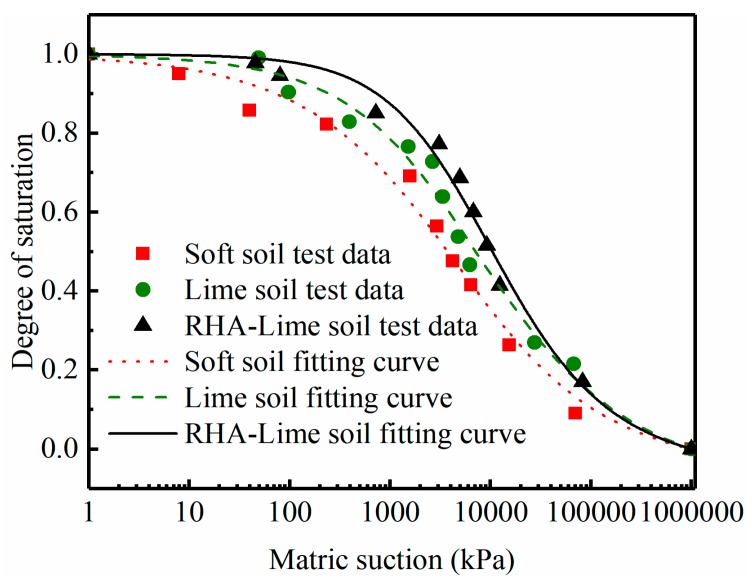
SWCC (soil–water characteristic curve) of the selected three kinds of soils.

**Figure 10 materials-12-03873-f010:**
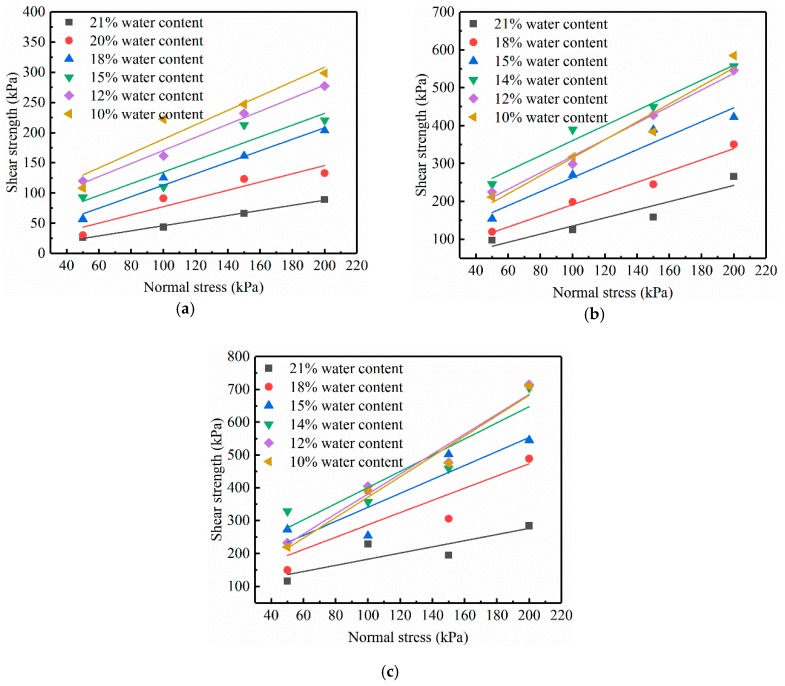
Shear strength test curve. (**a**) Soft soil; (**b**) Lime-stabilized soil; (**c**) RHA–lime stabilized soil.

**Figure 11 materials-12-03873-f011:**
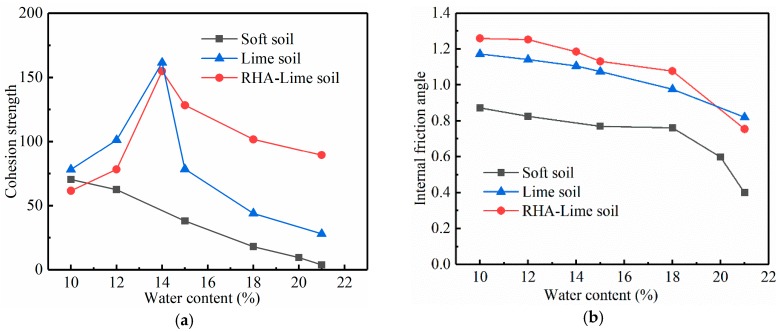
Relationship between shear strength indexes and water content. (**a**) Cohesion strength; (**b**) Internal friction angle.

**Figure 12 materials-12-03873-f012:**
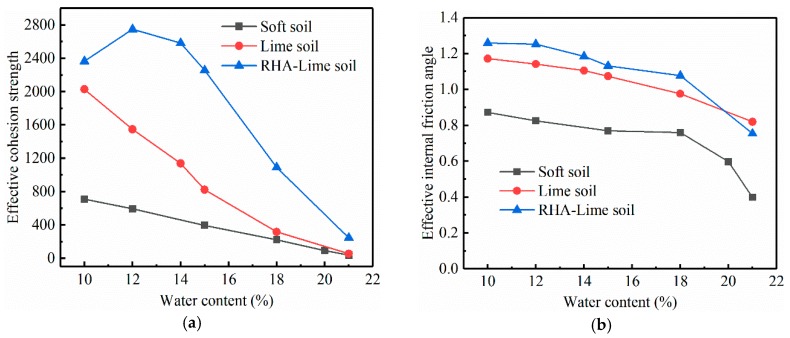
Relationship between effective shear strength index and water content. (**a**) Effective cohesion strength; (**b**) Effective internal friction angle.

**Figure 13 materials-12-03873-f013:**
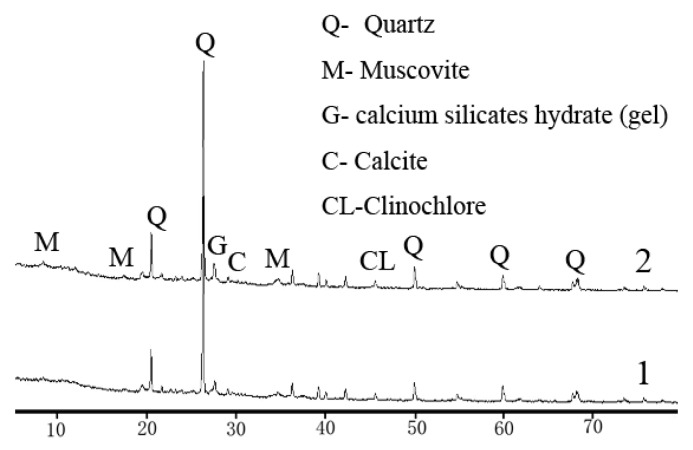
X ray diffraction pattern of stabilized soils.

**Figure 14 materials-12-03873-f014:**
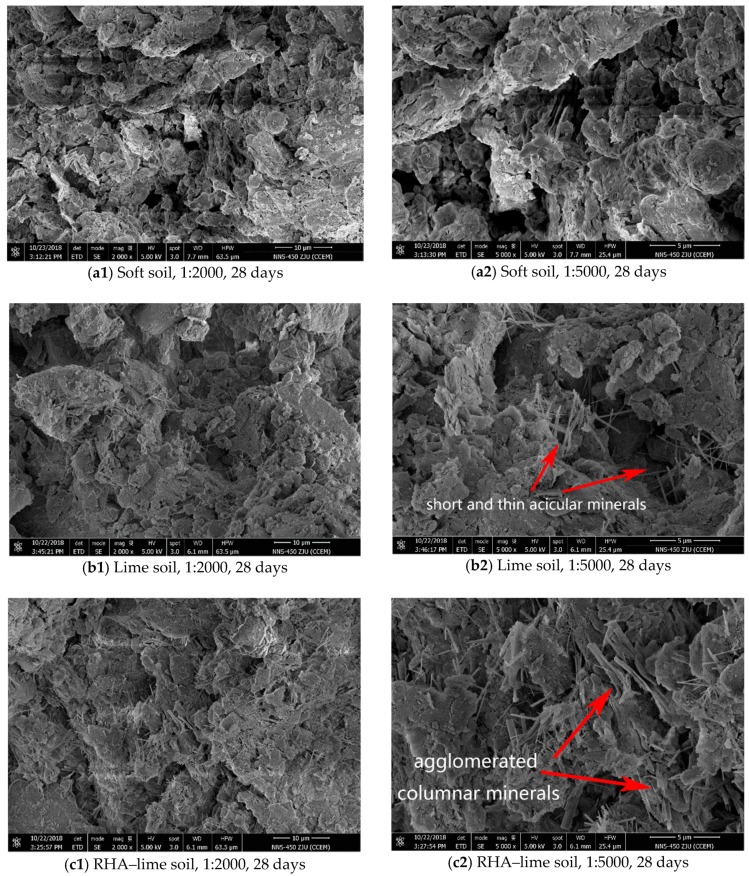
SEM images of three type soils under 28 day: (**a**) SEM images of soft soil under 28 day curing ages; (**b**)SEM images of Lime soil under 28 day curing ages; (**c**) SEM images of RHA-lime soil under 28 day curing ages.

**Figure 15 materials-12-03873-f015:**
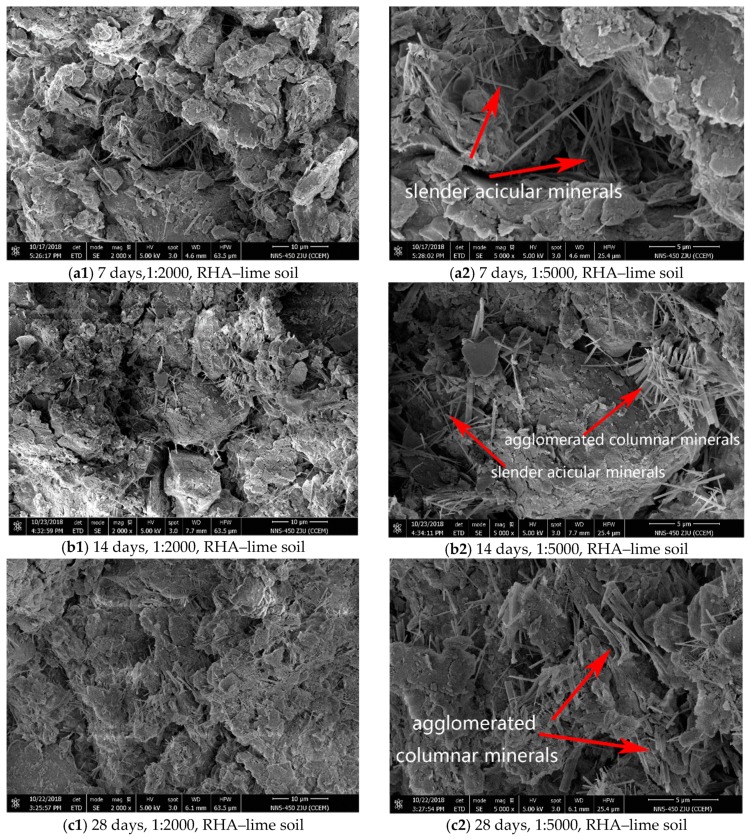
SEM images of RHA–lime stabilized soil under different curing ages: (**a**) SEM images of RHA-lime soil under 7 day curing ages; (**b**) SEM images of RHA-lime soil under 14 day curing ages; (**c**) SEM images of RHA-lime soil under 28 day curing ages.

**Table 1 materials-12-03873-t001:** Basic properties of soil.

Soil Type	Liquid Limit, WL/%	Plastic Limit, WP/%	Plastic Index, IP	Optimum Moisture Content/%	Maximum Dry Density/ (g/cm^3^)
Silt clay	38	19	19	18	1.798

**Table 2 materials-12-03873-t002:** Chemical composition of stabilized materials (%).

Materials	SiO_2_	CaO	Al_2_O_3_	MgO	Others
Lime	-	86.2	-	0.68	-
Rice husk ash	88.09	0.98	1.25	0.34	-

**Table 3 materials-12-03873-t003:** Results of the SWCC model coefficients.

	$h_{r}$	$a_{f}(\mathrm{AEV})$	$b_{f}$	$c_{f}$
Soft soil	3000	2000	0.509	1.581
Lime soil	3000	3059	0.589	1.192
RHA–lime soil	3000	7980	0.787	1.241

**Table 4 materials-12-03873-t004:** Results of the methylene blue test.

Soil Sample	*V_cc_* (mL)	MBV (g/100 g)	SSA (m^2^/g)
Soft soil	37.00	1.23	30.17
Lime-stabilized soil	32.00	1.07	26.10
RHA–lime stabilized soil	29.00	0.97	23.65

**Table 5 materials-12-03873-t005:** Soil expansion characteristics judgment table.

MBV(g/100g)	Degree of Expansion
0–4	Low
4–8	Medium
8–15	High
>15	Very high

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
