# Peer review of "Analysis of Strength Development and Soil–Water Characteristics of Rice Husk Ash–Lime Stabilized Soft Soil"

_materials, 2019, doi:10.3390/ma12233873_

Round 1

Reviewer 1 Report

The English language and the structure of the sentences require to be checked.

Overall, the manuscript could be written in a more concise form, especially in the methodology and description of the test procedures.

The authors may try to make the abstract more concise as it is currently very long and too detailed in terms of the results.

Page 2, line 53: the term "at home and abroad" might be revised as this is an international publication. Also, the next sentence should be revised to be more clear.

The dimensions of 50 mm in diameter and 50 mm high for the unconfined compressive strength and splitting strength tests do not seem to be in line with the 1:2 H:V standard specimen ratio normally used for the tests. Would the authors clarify the standard followed in this regard?

There is no need for the numbering in the conclusion section. Also, this section should be more descriptive and not only reparation of the previous numerical results.

Reviewer 2 Report

The manuscript has the form of a research report with good SEM analysis, on the basis of which the authors proposed conclusions. While preparing for the review of the manuscript, the reviewer reviewed the literature attached by the authors and in this sense he feels a bit unsatisfied. Authors should relate the results of their research and analysis to other authors' research carried out in this topic. It's about comparing results values and not overall conclusions.  The authors have certainly put a lot of work into the research, which, however, is not subject to evaluation.

The basic problem of the work is insufficient characterization of the soil used for testing. Why the authors do not provide the mineral composition of the soil in the description of the materials? It seems that the mineral (chemical) composition will be fundamental to the results. The soil containing clay will have different properties depending on what clay minerals are in it. The reviewer think that for silty clay with a different mineral composition, from another part of the world, and even from another region of China, the results obtained would be different. Reviewer is surprised that the authors state how much CaO, MgO was in rice, while it is not known whether Chinese soil did not contain, for example, carbonates.

Are the “Basic propertis of soil samples” (Table 1) results of the Authors' tests? The Authors write in Table 1 that the optimum moisture content of Chinese soil was 18%. However, it is not given by what method this OMC value was determined. It is important if other scientists, e.g. from India (where a lot of rice is grown) would like to compare the authors' results with their own. OMC will depend on the method of sample concentration, which is different in different regions of the world.
The title of the article suggested that rice husk ash-lime was added to many different soils, not one, in addition without giving its chemical composition. It is not even known if this composition contained swelling clay minerals or non-swelling clay minerals.

In the introduction, it is worth completing the review of the current state of knowledge. The reviewer proposes to present the results of other scientists in order to later try to relate them to Authors own. The introduction is very general, it will introduce a person who has not heard of rice husk ash-lime as a soil stabilizer. However, for researchers conducting research in this topic, showing other results in the Introduction and comparing the results presented in the manuscript (in discussion part) seems crucial.

Minor remarks:

Reviewers do not like the general rule of adding citations to obvious statements, probably aimed at ensuring that the publication has a good number of references. The selection of sources is good, but it contains errors - e.g. in my opinion the quotation between lines 63-65 is not from reference [4]. Do you have to give references to obvious sentences like in line 44-46?

While short names are desirable in main part of the manuscript, I suggest that authors do not use short names in their conclusions. For example, conclusion “(ii)” contains as many as four abbreviations  (RHA, AEV, AVE and SWCC). One of them is again explained air entry value (AEV), although previously introduced in the text. I propose to write all conculsions without abbreviations.

While the information that China is the largest producer of rice in the world is obvious, the amount (200 million tonnes) given is disputable. The given source [10] is in Chinese. Could the authors bring up other English-language ones? The authors continue to calculate how much rice hulls per year China alone produces. In general, the authors too often and unnecessarily emphasize in the article that the soil is Chinese, a lot of rice is in China, etc. After all, the idea of stabilizing the soil for rice husk ash-lime may also be of interest to readers from America, India, etc.

The text requires proofreading. For example, a sentence beginning with "And" appears several times in the text. The article is formatted with large spaces other than those in the journal templete.  I suggest changing the place of the legend in Fig. 9. and increasing the spacing on the horizontal axis.

Round 2

Reviewer 2 Report

Thank you for replying to the review.
Authors have added X-ray diffraciton pattern of soil used. This was one of the two key objections to the results presented. The question is, is the X-ray diffraction really for the same soil that was used in the work? I have my doubts. How did the authors carry out this, since the soil used to prepare samples has already been mixed with additives? The second complaint regarding the manuscript has unfortunately not been resolved. I still think that the manuscript has the form of a research report not an article. I think that authors should relate the results of their research and analysis to other authors' research carried out in this topic. I find the introduction very poor. I also miss the chapter devoted to discussing the results, among others, because no other authors’ results values were added in the Introduction chapter. In conclusion, I do not change my mind about the research report I review. The authors' research is a good basis for articles published in reputable journals such as this, but in my opinion the manuscript still needs improvement
